# Injectable pH and Thermo-Responsive Hydrogel Scaffold with Enhanced Osteogenic Differentiation of Preosteoblasts for Bone Regeneration

**DOI:** 10.3390/pharmaceutics15092270

**Published:** 2023-09-02

**Authors:** Jasmine L. King, Roopali Shrivastava, Pooja D. Shah, Panita Maturavongsadit, Soumya Rahima Benhabbour

**Affiliations:** 1Division of Pharmacoengineering and Molecular Pharmaceutics, UNC Eshelman School of Pharmacy, University of North Carolina at Chapel Hill, Chapel Hill, NC 27599, USA; jking116@live.unc.edu; 2Joint Department of Biomedical Engineering, North Carolina State University and The University of North Carolina at Chapel Hill, Chapel Hill, NC 27599, USA; roopalis@email.unc.edu (R.S.); pdshah1@email.unc.edu (P.D.S.); panita.maturavong@gmail.com (P.M.)

**Keywords:** chitosan-based hydrogel, cellulose nanocrystals, injectable hydrogels, osteogenic differentiation, osteogenesis, bone regeneration

## Abstract

Bone fractures are common in the geriatric population and pose a great economic burden worldwide. While traditional methods for repairing bone defects have primarily been autografts, there are several drawbacks limiting its use. Bone graft substitutes have been used as alternative strategies to improve bone healing. However, there remain several impediments to achieving the desired healing outcomes. Injectable hydrogels have become attractive scaffold materials for bone regeneration, given their high performance in filling irregularly sized bone defects and their ability to encapsulate cells and bioactive molecules and mimic the native ECM of bone. We investigated the use of an injectable chitosan-based hydrogel scaffold to promote the differentiation of preosteoblasts in vitro. The hydrogels were characterized by evaluating cell homogeneity, cell viability, rheological and mechanical properties, and differentiation ability of preosteoblasts in hydrogel scaffolds. Cell-laden hydrogel scaffolds exhibited shear thinning behavior and the ability to maintain shape fidelity after injection. The CNC-CS hydrogels exhibited higher mechanical strength and significantly upregulated the osteogenic activity and differentiation of preosteoblasts, as shown by ALP activity assays and histological analysis of hydrogel scaffolds. These results suggest that this injectable hydrogel is suitable for cell survival, can promote osteogenic differentiation of preosteoblasts, and structurally support new bone growth.

## 1. Introduction

Bone fractures are a major global public health concern and pose a significant economic burden on the healthcare system, with an increasing incidence in the elderly population. In the United States, it is estimated to cost $5 billion annually to treat bone defects [1]. Common disorders associated with complex or compromised bone fractures are diabetes, osteoporosis, bone tumors, endocrine and hormonal-related disorders, and other degenerative bone diseases. These fractures often fail to heal properly, requiring surgical intervention to facilitate bone repair. Bone grafting has seen increasing use in surgery, with currently more than two million procedures per year worldwide to facilitate bone regeneration for unrepaired bone defects [2]. There are several factors to consider when designing an ideal bone graft: defect size, tissue viability, cost, biomechanical characteristics, and the like [3]. The current gold standard treatment for repairing bone defects is autografts. Autografts possess osteogenic characteristics that promote bone healing, growth, and remodeling [4]. There are several limitations associated with autografts; however, the main limitations are donor-site morbidity, injury during harvesting, and remodeling issues of the implanted bone [3,5,6,7,8].

Bone graft substitutes are alternative approaches that have shown promise in providing structural benefits for bone regeneration. Bone substitutes consist of synthetic and/or natural biomaterials with or without incorporated biological factors. The ideal criteria to consider for engineering bone substitutes to improve clinical outcomes are selecting material that is biocompatible, biodegradable, osteoinductive, osteoconductive, cost-effective, and structurally similar to bone [3,4]. Injectable in-situ forming bone substitutes, such as calcium sulfate (CS) and calcium phosphate cement (CPC), are attractive candidates to fill irregularly shaped bone defects [3,9,10]. However, there are several drawbacks associated with these injectable materials. Calcium sulfate bone substitutes have limited ability to achieve optimal bone regeneration due to their fast degradation rates and weak internal mechanical strength, limiting their application to smaller bone defects [3,9]. Similar to calcium sulfate, calcium phosphate cement possesses unpredictable degradation kinetics and low flexural strength after the load is applied [3,9]. Given these concerns, injectable, in-situ forming, three-dimensional, porous hydrogel scaffolds are alternatives to these biomaterials.

Injectable hydrogels are well-suited biomaterials for tissue engineering applications due to their ability to incorporate bioactive molecule cells, fill any irregular shape or critically sized defect, and their minimally invasive and biodegradable nature, thus eliminating the need for removal [11,12]. In particular, chitosan-based hydrogels are attractive injectable biomaterials given their biocompatibility, biodegradability, unique antimicrobial properties, and ability to mimic native extracellular matrix (ECM) composition [11,13,14]. When exposed to an aqueous environment, CS exhibits shear thinning and self-healing properties, making it easily injectable. Additionally, CS-based hydrogels can be designed to exhibit fast gelation properties upon injection under physiological conditions. An FDA-approved CS-based scaffold, BST-CarGel^®^, has been shown to be effective at repairing cartilage [15]. This bioscaffold technology is mixed with the patient’s whole blood before implantation and injected into damaged cartilage for repair.

Furthermore, CS can be used as a bioscaffold material to support attachment and proliferation of osteoblasts for bone repair [13]. In addition, cellulose nanomaterials have shown promise as biomaterials for regenerative medicine applications owing to their biocompatibility, biodegradability, and high mechanical strength [16,17]. They have been widely studied as reinforcing agents for polymers. Various sources and isolation methods are available to synthesize nanocellulose [17]. However, the method and source of synthesis influence the size, dimensions and surface functionality of nanocellulose [17]. Given the versatility in fabrication methods, the mechanical and rheological properties of these nanomaterials are highly tunable, making them an attractive material to incorporate for bone regeneration.

In this study, we investigated the use of an injectable in-situ forming chitosan-based hydrogel to support the proliferation and differentiation of pre-osteoblasts for bone regeneration. Given the poor mechanical properties of chitosan-based scaffolds [13,14], we have incorporated cellulose nanocrystals (CNCs) as a nanomaterial to strengthen mechanical properties and support load-bearing implantation areas. We investigated the effects of CNCs on the rheological properties and cytocompatibility of the injectable cell-laden hydrogels. Moreover, we investigated the effects of cell seeding density on osteogenic differentiation potential. Hydrogels’ storage modulus, yield stress and viscosity properties were determined to assess the impact of CNCs and cells on the rheological properties of cell-laden hydrogels.

Additionally, cell viability was determined for a range of cell densities to determine optimal cell seeding density within the hydrogel system. The impact of hydrogel formulations on cell differentiation was also evaluated using an alkaline phosphatase activity assay corroborated with collagen and calcium histological staining. These results provide a strong foundation for developing our injectable cell-laden hydrogel technology and determining the required optimal cells and CNCs content that can result in optimal efficacy in vivo and effectively promote bone tissue regeneration.

## 2. Materials and Methods

### 2.1. Materials

Chitosan (CS) powder (85% deacetylated, 200–800 cP, Lot #: BCCC6283, 3 wt% in 0.1 M acetic acid), β-glycerophosphate (BGP, Lot #: SLCM7561), hydroxyethyl cellulose (HEC, MW: 90,000 Da, Lot #: MKCM0782), laboratory-grade Triton X-100, ascorbic acid-2-phosphate, and dexamethasone were purchased from Sigma-Aldrich (St. Louis, MO, USA). Luer-Lock (1 mL) syringes were purchased from Becton and Dickinson (Macquarie Park, Australia). Luer-Lock connectors were purchased from Baxter (Deerfield, IL, USA). BD PrecisionGlide 18 G × 1” hypodermic needles were purchased from Becton and Dickinson (Macquarie Park, Australia). Fisherbrand™ Disposable Base Molds were purchased from Fisher Scientific (Pittsburgh, PA, USA). Tissue-Tek^®^ Embedding Rings were purchased from Electron Microscopy Sciences (Hatfield, PA, USA). Alkaline Phosphatase (ALP) Activity Colorimetric Assay kit was purchased from Biovision, Inc. (Waltham, MA, USA). LIVE/DEAD^®^ Viability/Cytotoxicity kit containing Calcein AM and Ethidium Homodimer-1 reagents and Quant-iT™ PicoGreen™ dsDNA Assay kit and dsDNA reagents were purchased from ThermoFisher Scientific (Waltham, MA, USA). MC3T3-E1 subclone 4 preosteoblast cells were purchased from the American Type Culture Collection (ATCC; Manassas, VA, USA). Cell culture media was prepared using Dulbecco’s Modified Eagle Medium (DMEM; Gibco, Waltham, MA, USA) supplemented with 10% fetal bovine serum (FBS; Gibco, Waltham, MA, USA), 1% penicillin/streptomycin (P/S; Gibco, Waltham, MA, USA) and 4 mM L-glutamine (referred to as complete media). Osteogenic media was prepared using Dulbecco’s Modified Eagle Medium (DMEM; Gibco, Waltham, MA, USA) supplemented with 10% fetal bovine serum (FBS; Gibco, Waltham, MA, USA), 1% penicillin/streptomycin (P/S; Gibco, Waltham, MA, USA), 4 mM L-glutamine, 50 µg/mL ascorbic acid 2-phosphate, and 100 nM dexamethasone.

### 2.2. Preparation of Injectable Hydrogels

A 3% *w/v* chitosan (CS) stock solution was prepared as previously reported by stirring chitosan powder in 0.1 M acetic acid in deionized water at room temperature (at 25 °C) for 48 h [18,19]. The 1 M β-glycerophosphate (BGP) stock solution was prepared by dissolving BGP in deionized water. Hydroxyethyl cellulose (HEC) stock solution (25 mg/mL) was prepared by dissolving HEC in serum-free DMEM. In brief, as previously reported, the injectable thermogelling hydrogels were prepared using a three-component, two-step mixing process under aseptic conditions. The three components consisted of: (1) CS, (2) BGP, (3) HEC, cellulose nanocrystals (CNCs), and with or without preosteoblast cells in DMEM. The components were homogenously mixed using a Luer-lock connector to create a pre-hydrogel mixture for injection. 

### 2.3. Rheological Properties of Cell-Laden CNC-CS Hydrogels

Rheological characterization of cell-free and cell-laden CS and CNC-CS hydrogels was carried out using a rotational rheometer, Discovery Hybrid Rheometer (DHR-3), following hydrogel preparation. All rheology measurements were made using an 8 mm Sandblasted Peltier Plate. The samples were placed onto the lower plate surface, and the upper plate was lowered to a 0.5 mm gap distance. All viscosity measurements were performed using a logarithmic sweep and a shear rate of 0.1 to 100 s^−1^ at room temperature 25 °C). A continuous oscillation with a direct strain of 0.1–2000% and a constant frequency of 1 Hz was performed for all measurements at room temperature (at 25 °C) to determine yield stress. The point at which the storage modulus (G′) and loss modulus (G″) intersect was determined to be the yield stress. The complex viscosity was also determined at a constant frequency of 1 Hz. Viscoelasticity measurements were performed using dynamic frequency sweep testing. Before analysis, hydrogels were prepared and stored in 1X PBS at 37 °C for 24 h. For all measurements, the hydrogels were exposed to a constant strain amplitude of 1% and frequency of 0.1–20 rad/s. Gelation times were determined at 37 °C using the Peltier Plate heating system. The linear viscoelastic region of CS and CNC-CS hydrogels was previously optimized and reported [18,19]. All gelation measurements were performed using a dynamic time sweep at a constant strain and angular frequency: 0.05% and 20 rad/s.

### 2.4. In Vitro Cell Viability

To determine the cell viability of MC3T3-E1 preosteoblasts, 5 × 10^6^–2 × 10^7^ cells were encapsulated in 1 mL of pre-hydrogel. Cell-laden pre-hydrogels were seeded into an 8-well Nunc-chamber slide glass slide (50 µL; *n* = 3) and placed in an incubator at 37 °C/5% CO_2_ to allow gelation. After gelation, complete media was added to all samples. Twenty-four hours (24 h) following incubation, cell viability was evaluated using the LIVE/DEAD^®^ Viability/Cytotoxicity kit. The samples were imaged using a laser scanning confocal microscope 780 (LSM 780; UNC Neuroscience Microscopy Core, Chapel Hill, NC, USA) at 10× magnification. Twelve z-stacks were collected per sample.

### 2.5. In Vitro Cell Proliferation

CS and CNC-CS hydrogels were prepared as previously described with a cell density 5 × 10^6^ MC3T3-E1 preosteoblast cells per 1 mL of pre-hydrogel (100 µL; *n* = 6) and incubated at 37 °C/5% CO_2_ to allow gelation to take place. The hydrogel samples were seeded into a 6-well plate containing osteogenic media. On days 7, 14, and 21 hydrogel scaffolds were lysed, and DNA concentration was assessed using Quant-iT™ PicoGreen™ dsDNA Assay kit. Fluorescence was measured using a fluorescence microplate reader.

### 2.6. In Vitro Osteogenic Differentiation of Cell-Laden Hydrogel Scaffolds

Hydrogels containing 5 × 10^6^ MC3T3-E1 preosteoblast cells were prepared and seeded into 6-well plates with osteogenic media. Osteogenic media was replaced every two days for up to 21 days. On days 7, 14, and 21, hydrogel scaffolds were harvested and analyzed to determine alkaline phosphatase activity, collagen formation, and calcium deposition.

### 2.7. Alkaline Phosphatase Activity

In brief, 1 mL of 1% Triton X-100 solution was added to each hydrogel sample and the cells were lysed by performing three repeat freeze-thaw cycles. To measure intracellular ALP activity, 20 mL of cell lysate and 60 mL of ALP assay buffer were added to a 96-well plate. Subsequently, 50 mL of 5 mM *p*-nitrophenyl phosphate was added to each well containing hydrogel samples. ALP standards were prepared and added to the 96-well plate according to manufacturer protocol (Alkaline Phosphatase Activity Colorimetric Assay kit; BioVision, Waltham, MA, USA). The samples were incubated in the dark at 25 °C for 60 min to allow the reaction to take place. ALP activity was determined by measuring optical density at 405 nm using a Biotek Synergy 2 microplate reader (Santa Clara, CA, USA).

### 2.8. Histological Staining and Imaging

Cell-laden hydrogels were prepared and seeded into 6-well plates containing osteogenic media as previously described. On days 7, 14, and 21, hydrogel samples were placed in an equal volume of 4% paraformaldehyde (PFA) fixative and incubated for 30 min at room temperature (at 25 °C) in the dark. Following incubation, samples were gently dehydrated by immersing the samples in a graded series of sucrose solutions ranging from 10–30% sucrose in 1X PBS. Hydrogel samples were subsequently embedded in cryomolds using optimal cutting temperature (OCT) embedding media and simultaneously frozen using dry ice. The hydrogel blocks were then fixed onto the cryostat base for sectioning. The samples were subsequently cut into cryosections with a thickness of 10 μm and stained with hematoxylin and eosin (H&E) to determine ECM composition and with von Kossa to detect the presence of calcium deposits. All hydrogel sections were imaged using a Nikon Ti2 Eclipse Color and Widefield Microscope (UNC Neuroscience Microscopy Core, Chapel Hill, NC, USA) at 20× magnification.

### 2.9. Statistical Analysis

Statistical analysis was performed using one- and two-way Tukey’s multiple comparisons test with GraphPad Prism Software v9.3.1.

## 3. Results

### 3.1. Injectable CNC-Hybridized CS Hydrogels

The CNC-hybridized chitosan-based injectable hydrogels were prepared as previously reported [18]. The concentration of each component in the hydrogel formulation has been previously optimized and reported in the literature (Figure 1A). The optimized hydrogel formulations used in this study are shown in Table 1. The concentration of the primary and secondary gelling agents, BGP and HEC, was optimized to maintain cell survival at different seeding densities and promote faster gelation kinetics [18]. BGP, a weak base, acts as a neutralizing agent that increases the pH (>6.2) of the CS solution and reduces the electrostatic repulsions between the CS-CS molecule. As the temperature increases, the presence of BGP allows for hydrophobic interactions between the CS-CS-CS chains to predominate, resulting in the formation of a gel [20,21]. The complete mechanism of formation has been confirmed via Fourier-transform infrared (FTIR) spectroscopy and previously reported in the literature [17]. In brief, at room temperature (25 °C), non-covalent crosslinking, via ionic interactions and hydrogen bonding, is the predominate interaction within the polymer-polymer backbone owing to its liquid-like state. Under physiological conditions, pH 7.4 and 37 °C, primary non-covalent interactions, i.e., electrostatic, hydrophobic, and hydrogen bonding interactions, in addition to secondary chemical crosslinking via a Schiff base reaction between the glyoxal molecules in HEC and the amine groups in the CS network predominate thus promoting fast gelation (Figure 1B). As a result, these hydrogels can be injected intraosseously to form an in-situ gel at the critical defect site (Figure 1C).

### 3.2. Rheological Properties of Injectable CNC-Hybridized CS Hydrogels

Previous work has demonstrated that the incorporation of CNCs at 1.5% was able to enhance the hydrogel’s mechanical properties and promote osteogenic differentiation of MC3T3-E1 in 3D bioprinted hydrogel scaffolds (referred to as bioinks) [22]. Therefore, in this study, we set out to evaluate the cell viability of MC3T3-E1 cells at varying cell seeding densities, the rheological effects of incorporating MC3T3-E1 cells, and the osteogenic differentiation potential of the injectable hydrogels with and without 1.5% CNCs. The initial screening process of CS and CNC-CS hydrogels was performed with varying MC3T3-E1 cell loading densities ranging from 5 to 20 million. Results demonstrated that higher cell loading densities ≥ 10 million cells per mL hydrogel were viable following encapsulation, as shown in Appendix A. However, these cell concentrations were not uniformly distributed throughout the pre-hydrogel mixture, resulting in a non-homogenous cell-laden hydrogel formulation (Appendix A). Given these results, cell concentrations ≥ 10 million cells per mL hydrogel were not further characterized in this study.

Rheological measurements were evaluated on all optimized cell-laden CS and CNC-CS hydrogel scaffolds to determine viscosity, yield stress, and gelation kinetics. These properties are significant parameters that influence the hydrogels’ injectability, ability to maintain shape fidelity, and ability to retain encapsulated cells at the defect site upon injection. Ideally, hydrogels that exhibit (1) high viscosity to maintain shape and prevent hydrogel escape at the site of injection, (2) shear-thinning behavior during injection, and (3) fast gelation kinetics to retain cells and prevent loss of hydrogel following injection [23].

The viscosity of CS and CNC-CS hydrogels was investigated to determine the effect of incorporating MC3T3-E1 cells on hydrogel flow behavior. The viscosity measurements were determined using a flow sweep analysis (Figure 2A). Results demonstrated that all hydrogel formulations exhibited shear-thinning properties as the shear rate increased, demonstrating their injectability. Furthermore, there was no statistically significant difference in viscosity between cell-free and cell-laden injectable hydrogels. However, at a frequency of 1 Hz, results demonstrated that there was statistically significant difference between cell-free CS and CNC-CS hydrogels (23.12 ± 1.285 vs. 34.11 ± 2.560; *p* < 0.0001), cell free CS and cell-laden CS hydrogels (23.12 ± 1.285 vs. 25.34 ± 0.9209; *p* < 0.05), cell-laden CS and CNC-CS hydrogels (25.34 ± 0.9209 vs. 30.35 ± 1.038; *p* < 0.0001), respectively (Figure 2B). The differences observed in cell free and cell-laden CNC-CS hydrogels can be attributed to cells occupying void space within the polymer matrix, thus causing less interactions between the polymer backbone, which ultimately may result in a decrease in the complex viscosity.

The yield stress of injectable hydrogel formulations was measured to characterize further the ability of hydrogels to maintain their shape fidelity upon injection. Yield stress can be defined as the point at which the hydrogel begins to flow under applied stress. No statistically significant differences were observed in both cell free hydrogel scaffolds, i.e., with and without CNCs (*p* = 0.50). Conversely, results showed that in the presence of cells, CNC-CS hydrogels exhibited a significant increase in yield stress compared to CS hydrogels (610.5 ± 109.0 vs. 833.5 ± 34.70; *p* < 0.05). More importantly, incorporating cells into either hydrogel formulations (with and without CNCs) did not significantly change the hydrogels’ yield stress (Figure 2C).

The storage modulus was evaluated to determine the elastic behavior or stiffness of the hydrogel scaffolds. Results from the mechanical testing, shown in Figure 2D, demonstrate that the addition of CNCs and cells increases the mechanical properties of CS hydrogels. Moreover, cell-laden CNC-CS hydrogels exhibited significantly higher modulus in comparison to cell free (6713 ± 993.6 vs. 12,698 ± 247.13; *p* < 0.005) and cell-laden CS hydrogels (8650 ± 494.9 vs. 12,698 ± 247.13; *p* < 0.05). These results indicate that the CNC-CS hydrogel formulation can achieve desirable mechanical stiffness to mechanically induce osteogenic differentiation of preosteoblast [24].

As previously discussed, it is important to maintain the shape fidelity of the hydrogels upon injection and retain the encapsulated cells at the defect site. To characterize the gelation time of cell free and cell-laden hydrogel formulations, gelation kinetics were analyzed. Given the thermogelling behavior of the injectable hydrogels, the gelation kinetics were determined by performing a dynamic time sweep under constant temperature, strain, and angular frequency. Gelation is the crossover between the storage (G′) and loss modulus (G″). The gelation analysis showed that all hydrogel formulations gelled in less than 7 s, and hydrogels continued to stiffen 5 min after injection (Figure 3 and Appendix A). These results demonstrate that cells did not affect CS and CNC-CS thermogelling properties. Furthermore, the gelling behavior of cell free and cell-laden hydrogels suggests instantaneous gelation upon injection under physiological conditions. A detailed report of all rheological analyses is summarized in Table 2.

### 3.3. In Vitro Cell Viability of MC3T3-E1 Cells

Cell viability was assessed to determine the cytocompatibility of injectable hydrogels. MC3T3-E1 preosteoblast cells were encapsulated in CS and CNC-CS hydrogels at a cell loading density of 5 million cells per 1 mL of hydrogel. In previous studies, we have demonstrated the ability to encapsulate neural stem cells in injectable hydrogels for the treatment of glioblastoma and MC3T3-E1 cells in bioink formulations for tissue engineering applications [19,22]. Herein, we report the semi-quantitative analysis of MC3T3-E1 cell viability in the injectable CS and CNC-CS hydrogels using confocal microscopy. Live image analysis was used to detect viable vs. non-viable cells within the hydrogel scaffold following twenty-four hours of encapsulation. As illustrated in Figure 4, MC3T3-E1 cells showed similar cell viability in both hydrogel scaffolds. Cell viability was maintained over 21 days, as illustrated in Appendix A. Furthermore, incorporating CNCs enhanced and maintained the proliferation of preosteoblasts compared to CS hydrogels (Appendix A). These results indicate that the components within the injectable hydrogel do not significantly impact cell viability, thus suggesting these hydrogels are cytocompatible and suitable biomaterials.

### 3.4. In Vitro Osteogenic Differentiation of MC3T3-E1 Cells in Injectable Hydrogels

In normal conditions, bone tissue formation occurs in three stages: (1) cell proliferation, (2) matrix synthesis and maturation and (3) matrix mineralization [25,26]. Pre-osteoblasts are key regulators in matrix maturation and mineralization by expression of early markers such as alkaline phosphatase and osteocalcin, which are necessary for collagen production [25,26]. To evaluate the osteogenic differentiation potential of MC3T3-E1 cells in the CS and CNC-CS hydrogels, in vitro osteogenesis assays were performed. Preosteoblasts were encapsulated in the hydrogels cultured in osteogenic media, and ALP activity, collagen formation, and calcium deposition were investigated over 21 days. As shown in Figure 5A and Appendix A, within the first seven days of culturing, ALP activity was significantly increased in MC3T3-E1 cells in CNC-CS hydrogels in comparison to CS hydrogels (0.002 ± 0.0003 vs. 0.0008 ± 0.0001; *p* < 0.005) showing a faster onset of ALP activity in the presence of CNCs. At day 14, the ALP activity remained unchanged in the cell-laden CS-CNC hydrogels. In addition, on day 14, the ALP activity increased in the cell-laden CS hydrogels to reach similar activity observed in the CS-CNC hydrogels. A graphical representation of ALP activity for cell concentrations > 10 million cells per mL hydrogel is illustrated in Appendix A. Given the challenges with preparing these formulations, as shown in Appendix A, there was no correlation or additive effect between increasing the cell density and ALP activity. These findings suggest that incorporating CNC nanomaterials significantly upregulates the expression of early osteogenesis marker ALP. In bone, collagen is the primary structural component of the organic matrix (>90%) [27,28]. Similarly, calcium is the most abundant mineral in human bone and teeth [29]. Therefore, the differentiation of MC3T3-E1 cells was further assessed by semi-quantitatively examining the collagen formation in the extracellular matrix (ECM) and calcium deposition in hydrogel scaffolds, as shown in Figure 5B,C. The percent area of stained ECM and deposited calcium were analyzed, and results are shown in Figure 5D,E. Similarly, the formation of collagen (41.1 ± 0.511 vs. 22.9 ± 3.61 and 49.9 ± 7.93 vs. 30.4 ± 3.48; *p* < 0.05) and the deposition of calcium (3.12 ± 0.194 vs. 1.46 ± 0.603) in hydrogel scaffolds were significantly increased at day 14 and 21 in CNC-CS compared to CS-only hydrogels. These results demonstrate that both hydrogel scaffolds can promote osteogenic differentiation of preosteoblast. However, the addition of CNCs to CS hydrogels induces the early expression of markers for osteogenesis and further promotes cell survival, differentiation, matrix maturation and mineralization to support bone formation.

## 4. Discussion

Although autografts are effective in bone reconstruction, they are associated with severe complications that lead to graft failure and improper bone healing. Research efforts have focused on the development of innovative injectable biomaterials that closely mimic the extracellular matrix and architecture of bone tissue. Factors such as polymer type, concentration, crosslinking behavior, and/or mechanisms can influence a biomaterial’s ability to induce bone formation [24]. Hydrogels possess the ability to provide mechanical and physical support to promote cell growth, proliferation, and differentiation for bone healing. Due to their three-dimensional network, hydrogels provide an environment for the encapsulation of cells and bioactive molecules and are excellent structures for integration into bone tissue. Given these advantages, we investigated a novel injectable, in-situ CNC-CS-based hydrogel system to evaluate its potential as a scaffold to promote osteogenesis and differentiation of preosteoblasts in vitro.

The biodegradability and mechanical integrity of hydrogel scaffolds are two key properties that determine their ability to promote proper bone healing and optimal efficacy in vivo. In this study, CNCs were incorporated as a nanomaterial to improve rheological and mechanical properties. All hydrogel formulations investigated in this study exhibited shear-thinning properties with similar viscosity measurements under increasing shear rates. Moreover, complex viscosity measurements were comparable and within the standard range (25–4540 Pa·s) [30] for both cell free and cell-laden CS and CNC hydrogels, indicating their potential ability to resist shear forces within the tissue post-injection. The addition of CNCs to CS hydrogels had a significant impact on yield stress and storage modulus. CNC-CS hydrogels exhibited greater yield stress and storage modulus than CS hydrogels alone.

Moreover, the significantly higher modulus of CNC-CS hydrogels demonstrates their potential ability to mimic the extracellular matrix of bone in vivo. Additionally, in cell-laden hydrogels, the yield stress and storage modulus increased concomitantly in the presence of CNCs. This could lead to better shape fidelity and mechanical integrity when injected in vivo.

It is well known that osteogenic differentiation of cells within biomaterials is a requisite for effective bone tissue formation. Following a bone fracture, bone growth and development usually occur over 3 to 12 weeks [1]. Hence, ALP activity and ECM components were analyzed to determine the ability of CS and CNC-CS hydrogels to promote bone formation. The histological analysis showed that the cell-laden hydrogels promoted collagen formation and calcium deposition for up to 21 days, thereby supporting bone maturation and mineralization of the ECM. Similar to our previous observations with bioink formulations [19], results showed that incorporating CNCs within the hydrogel formulation promoted enhanced osteogenesis, which could potentially enhance efficacy for bone healing. It is worth noting that a longer culture period greater than three weeks could provide more detailed observation on the ability of CS and CNC-CS hydrogels to induce osteogenic differentiation of preosteoblasts and support bone maturation and mineralization.

## 5. Conclusions

In conclusion, this study provides insights into a novel cellulose nanocrystal hybridized chitosan-based injectable hydrogel platform suitable for cell encapsulation to enable the osteogenic differentiation of osteoblasts precursor cells to osteocyte-like cells. Our results demonstrate that the incorporation of CNC nanomaterial improved the rheological and mechanical properties of our CS-based injectable hydrogel system, making it an attractive cell delivery system for tissue engineering applications. In vitro histological analysis demonstrated the significant upregulation of osteocyte-like activity within seven days. Furthermore, CNC-CS hydrogels maintained the ability to induce bone maturation and mineralization over 21 days. Currently, in vivo studies are underway utilizing bone marrow-derived mesenchymal stem cells (BM-mMSCs) isolated from BALB/c mice to investigate the regenerative capacity of this cellulose nanocrystal hybridized chitosan-based injectable platform in a BALB/c Calvarial defect model. Additionally, research efforts will further investigate this injectable BM-mMSC-laden hydrogel system in a large animal model for repairing a critically-sized bone defect.

## 6. Patents

SRB and PM are inventors on a patent application related to this work filed by the University of North Carolina, Office of Technology Commercialization (UNC OTC) (PCT International Application PCT/US2019/034492). The authors declare no conflict of interest.

## Figures and Tables

**Figure 1 pharmaceutics-15-02270-f001:**
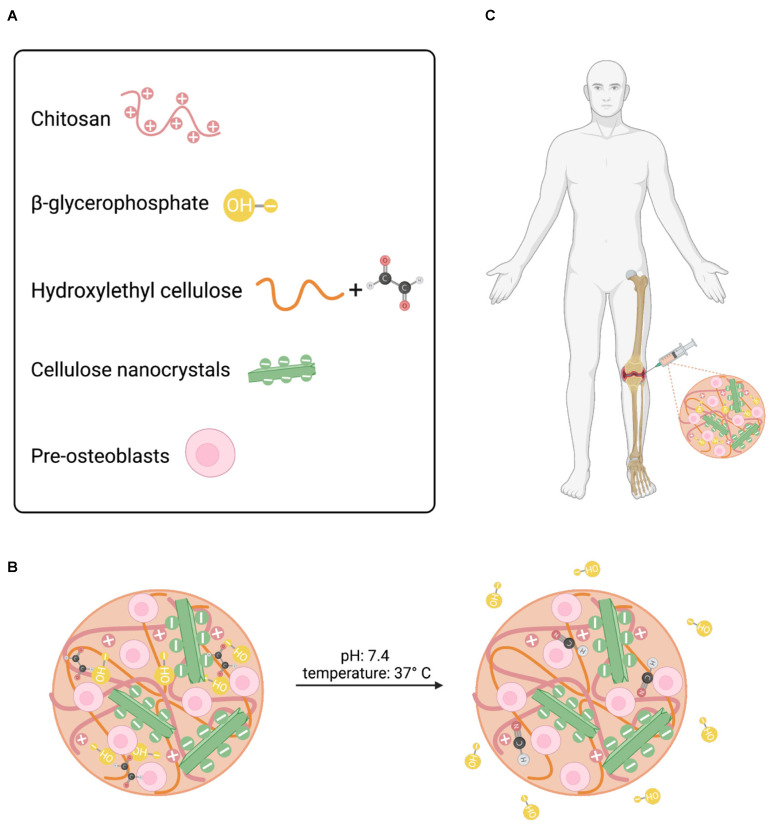
Schematic illustration of injectable hydrogel system. (**A**) Formulation components for injectable hydrogel system. (**B**) Crosslinking mechanisms under physiological conditions. (**C**) Desired site of injection.

**Figure 2 pharmaceutics-15-02270-f002:**
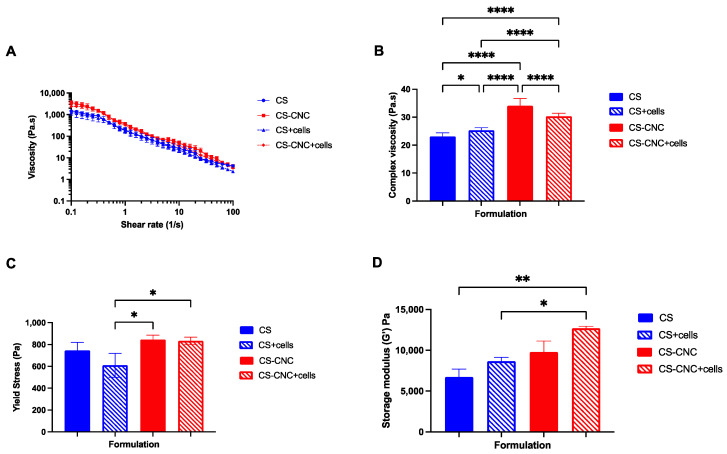
Rheological properties of CS and CNC-CS hydrogel with and without MC3T3-E1 cells. (**A**) Viscosity of injectable cell-laden hydrogels at a shear rate of 0.1–100 s^−1^. (**B**) Complex viscosity (* indicates *p* < 0.05 and **** indicates *p* < 0.0001) and (**C**) yield stress of injectable cell-laden hydrogels at a constant frequency of 1 Hz (* indicates *p* < 0.05; *n* = 3). (**D**) Storage modulus of injectable hydrogels at a constant strain amplitude of 1% and frequency of 0.1–20 rad/s (* indicates *p* < 0.05 and ** indicates *p* < 0.005; *n* = 3).

**Figure 3 pharmaceutics-15-02270-f003:**
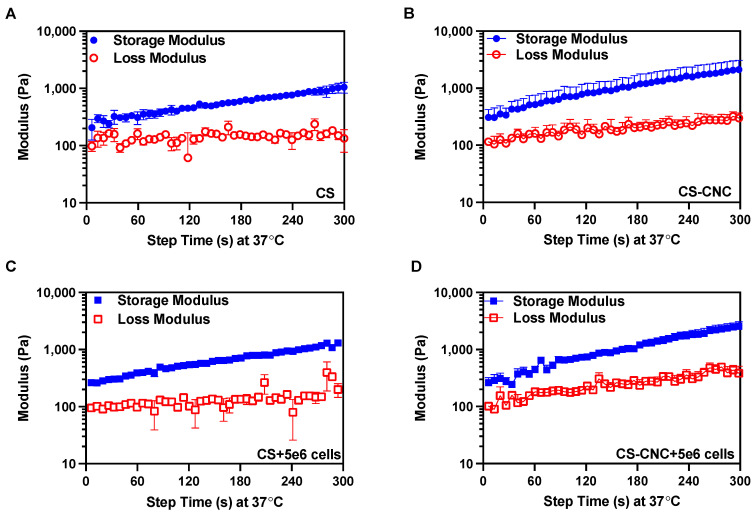
Gelation kinetics of CS and CNC-CS hydrogels with and without MC3T3-E1 cells. Gelation time of (**A**) CS only, (**B**) CS-CNC, (**C**) CS plus 5 × 10^6^ cells, and (**D**) CS-CNC plus 5 × 10^6^ cells determined by time at crossover G′ and G″ (*n* = 3).

**Figure 4 pharmaceutics-15-02270-f004:**
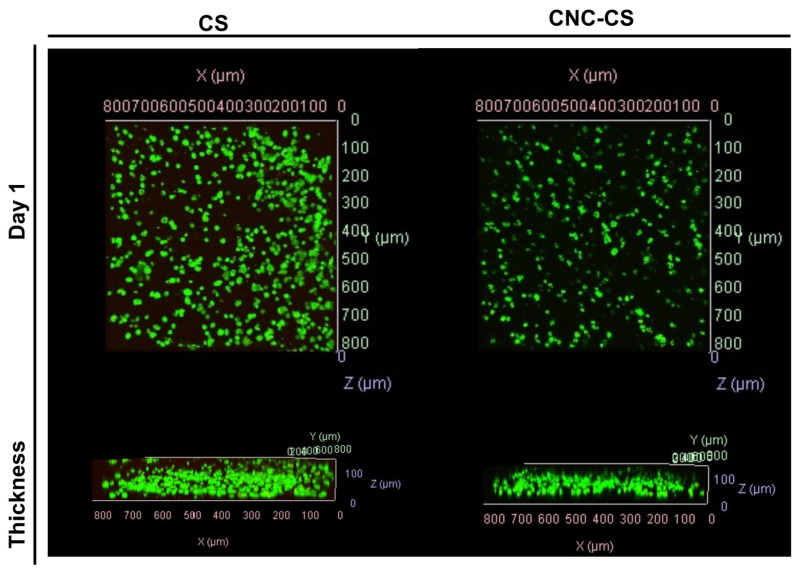
In vitro cell viability of MC3T3-E1 cells in injectable hydrogel scaffolds. Confocal fluorescent images of MC3T3-E1 cells (5 × 10^6^ cells/mL hydrogel) in CS and CNC-CS hydrogels (*n* = 3; 50 µL samples).

**Figure 5 pharmaceutics-15-02270-f005:**
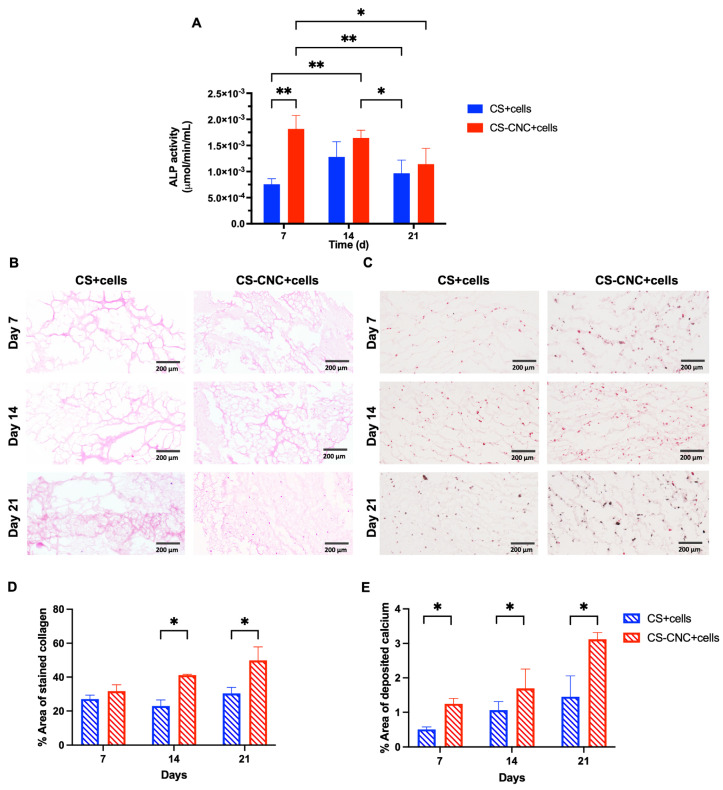
In vitro osteogenic differentiation of MC3T3-E1 cells in hydrogel scaffolds. (**A**) ALP activity of cell-laden CS and CNC-CS hydrogel over 21 days (* indicates *p* < 0.05 and ** indicates *p* < 0.005; *n* = 6; 100 μL samples). (**B**) H&E staining of newly formed ECM in cell-laden CS and CNC-CS hydrogels evaluated at days 7, 14 and 21 (*n* = 3; 100 μL samples). (**C**) Von Kossa staining of cell-laden CS and CNC-CS hydrogels over 21 days (*n* = 3; 100 μL samples). All scale bars represent 200 µm. (**D**) Percent area of newly formed ECM in CS and CNC-CS hydrogels at day 7, 14 and 21 post-incubation in osteogenic culture media quantified by ImageJ (version 1.52a). (**E**) Percent area of calcium deposition in CS and CNC-CS hydrogels at day 7, 14 and 21 post-incubation in osteogenic culture media quantified by ImageJ (version 1.52a).

**Table 1 pharmaceutics-15-02270-t001:** Injectable hydrogel formulations used in this study.

Formulation	CS (%*w/v*)	BGP (mM)	HEC (mg/mL)	CNCs (%*w/w*)	Cells (per mL Hydrogel)
1	2	100	0.5	0	0
2	2	100	0.5	1.5	0
3	2	100	0.5	0	5 × 10^6^
4	2	100	0.5	1.5	5 × 10^6^

**Table 2 pharmaceutics-15-02270-t002:** Summary table of injectable hydrogel rheological properties.

Rheological Properties	CS	CS + Cells	CNC-CS	CNC-CS + Cells
Complex viscosity (Pa·s) at 1 Hz	23.12 ± 1.285	25.34 ± 0.920	34.11 ± 2.560	30.35 ± 1.038
Yield stress (Pa)	746.5 ± 73.25	610.5 ± 109.0	846.3 ± 38.92	833.5 ± 34.70
Storage modulus (Pa)	6713 ± 993.6	8650 ± 494.9	9791 ± 1331	12,698 ± 247.13
Gelation time (s)	<7	<7	<7	<7

## Data Availability

The data presented in this study are within the article and Appendix A or on request from the corresponding authors.

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
