# Peer review of "Injectable pH and Thermo-Responsive Hydrogel Scaffold with Enhanced Osteogenic Differentiation of Preosteoblasts for Bone Regeneration"

_pharmaceutics, 2023, doi:10.3390/pharmaceutics15092270_

Round 1
Reviewer 1 Report (Previous Reviewer 3)
I can accept the revised manuscript.
Author Response
Many thanks.
Reviewer 2 Report (Previous Reviewer 2)
Asterisks are still absent in Figure 5
Can HE staining stain collagen? How is the semi-quantity result of collagen obtained?
Figure S4 is the proliferation data, not the viability after long-term culture.
Author Response
Response to reviewers’ comments:
Asterisks are still absent in Figure 5
Response: Thank you for noting the missing asterisks. We have added asterisks in Figure 5.
Can HE staining stain collagen? How is the semi-quantity result of collagen obtained?
Response: Yes, H&E can stain collagen blue; however, in our H&E staining we stained for the ECM (red/dark pink) which mainly consists of collagen. We have made this clarification in the text and Figure 5 caption to clarify this point in the revised manuscript.
Figure S4 is the proliferation data, not the viability after long-term culture.
Response: Although we agree that DNA count does not fully correlate with long-term cell viability, the ECM formation and calcium deposition data over 21 days (Figure 5) do correlate with long-term cell viability.
This manuscript is a resubmission of an earlier submission. The following is a list of the peer review reports and author responses from that submission.
Round 1
Reviewer 1 Report
In general, abstracts should be of a structured nature; authors often provide a substantial introduction, but lack proper methodology and fail to highlight the values of the main findings presented in the text to support the conclusion and perspective.
The introduction is well-presented; however, the material developed is not solely focused on chitosan. The authors do not address the advantages and disadvantages of using HEC and BGP in the final system, nor do they mention anything about which component of the formulation will possess the property to become a stimuli-responsive system and its respective application advantages and disadvantages.
The authors need to include batch numbers for all the raw materials used. The methodology, although not groundbreaking, should provide more detailed information for the interested scientific community to reproduce, particularly in the production of the system. For example, it should specify the proportion of the 3 components used and the agitation method and duration for each step.
"What is room temperature? 25°C? Please describe. The article to be published in this journal should contain an in vivo study demonstrating the activity, evaluating the system's inflammatory response, and the bone growth response. The cranial vault model would be suitable for the proposed study. However, in its current condition, due to the lack of novelty in the system, even though the authors demonstrate great intellectual ability to correlate the results and maintain high scientific standards, the study remains preliminary. Therefore, I recommend that the article should be published in a journal within the area of materials applied to dentistry, preferably in the MDPI group. I suggest transferring the submission to the 'DENTRISTRY JOURNAL'."
Author Response
We would like to thank the reviewer for reviewing our manuscript. We have provided a point-by-point response to address all your comment (see attached).
Regards,
Rahima

Reviewer 2 Report
This study presents an investigation of injectable hydrogel scaffold materials for bone tissue regeneration. The study aimed to assess the mechanical properties of injectable chitosan hydrogels, both with and without cell encapsulation. Additionally, the research evaluated cell viability and the effects on osteogenic differentiation of the hydrogels. ALP activity assays and histological staining results demonstrated that CNC-CS hydrogels significantly increased the osteogenic activity and differentiation of preosteoblasts. However, there are some issues that need to be addressed:
- While the hydrogel possesses temperature-responsive properties, the data does not confirm a solution-gel transition at a specific temperature.
- Although cell viability at day 1 was studied, it would be beneficial to investigate cell viability and cell proliferation after long-term culture.
- Figure 5 requires the inclusion of asterisks to indicate statistical differences.
- While H&E staining is useful for visualizing tissue structure, it may not be sufficient to determine the exact amount of collagen deposition.
- Given the lack of information on cell activity and proliferation, the ALP activities were assessed per hydrogel rather than being normalized by a single cell. And the latter represents the cell differentiation ability better.
Author Response

(The authors gave the same response as above.)

Reviewer 3 Report
I suggest a major revision of this study before publication:
1. A photo of the hydrogel before and after gelatin is suggested to add.
2. The title of this manuscript is "Injectable pH and thermo-responsive hydrogel scaffold with enhanced osteogenic differentiation of preosteoblasts for bone regeneration". Thus, the authors need to add the response parameters of pH and temperature for the hydrogel gelatin.
3. The injecting time should also be provided.
4. In Figure 5B and C, please add the scale bars.
5. If possible, animal experiment is suggested.
6. If possible , recent reference about injectable hydrogel is suggested for the Introduction <Jiaheng Liang, Kun Zhang, Jiankang Li, et al., Injectable Protocatechuic acid based composite hydrogel with hemostatic and antioxidant properties for skin regeneration, Materials & Design 222 (2022) 111109.>.
Author Response

(The authors gave the same response as above.)
